# Cross-Cultural Adaptation and Validation of the Person-Centered Therapeutic Relationship in Physiotherapy Scale to European Portuguese

**DOI:** 10.3390/healthcare12232455

**Published:** 2024-12-05

**Authors:** João Moreira, Lúcia Domingues, Margarida Silva, Carmen Caeiro

**Affiliations:** 1Department of Physical Medicine and Rehabilitation, Hospital Amato Lusitano, Unidade Local de Saúde de Castelo Branco, 6000-085 Castelo Branco, Portugal; 2Instituto Politécnico de Setúbal, Escola Superior de Saúde, 2914-503 Setúbal, Portugal; 3NOVA Medical School, Faculdade de Ciências Médicas, Universidade NOVA de Lisboa, 1169-056 Lisboa, Portugal; 4Comprehensive Health Research Centre, NOVA Medical School, Universidade NOVA de Lisboa, 1150-082 Lisboa, Portugal; 5Department of Physical Medicine and Rehabilitation, SAMS—Serviço de Assistência Médico-Social, do Mais Sindicato, 1070-128 Lisboa, Portugal

**Keywords:** cross-cultural adaptation, structural validity, internal consistency, therapeutic relationship, patient–provider communication, person-centered practice, physiotherapy

## Abstract

Background/Objectives: Implementing a person-centered practice is considered a priority in healthcare, and the development of a solid and positive therapeutic relationship is a fundamental element. This study conducted a cross-cultural adaptation of the Person-Centered Therapeutic Relationship in Physiotherapy Scale to European Portuguese and contributed to its validation through the study of its structural validity and internal consistency. Methods: This study was conducted from October 2021 to July 2023 and included two phases: a methodological study of cross-cultural adaptation (phase 1) and a transversal study to assess the psychometric properties (phase 2). Structural validity was analyzed using exploratory factor analysis and internal consistency was estimated using Cronbach’s alpha (α). A *p*-value < 0.05 was considered to indicate statistical significance. Results: The Person-Centered Therapeutic Relationship in Physiotherapy Scale was successfully culturally adapted to European Portuguese (phase 1). During phase 2, 203 individuals [mean age: 50.16 ± 13.10 years (range 18–80 years)] with musculoskeletal conditions, mostly female (63.1%), were recruited. The factorial solution explained 74.7% of the total variability and retained three factors, grouping items 9 to 15 in a common factor (professional empowerment and therapeutic communication). Adequate internal consistency was found (Cronbach’s α = 0.889). Conclusions: This study culturally adapted an instrument to European Portuguese, which allows the assessment of the person-centered therapeutic relationship in physiotherapy, presenting adequate internal consistency. Future studies should contribute to the remaining validation of the instrument so that it can be available to the Portuguese population.

## 1. Introduction

Musculoskeletal conditions comprise more than 150 different conditions and are typically characterized by pain, often persistent [1]. The 2019 Global Burden of Disease Study reported a prevalence of 1.71 billion people with musculoskeletal conditions and 149 million years of life lived with disability (YLDs) worldwide [2]. Within these conditions, low back pain was identified as the most prevalent [2,3,4]. Since 1990, the age-standardized prevalence and YLDs rates showed modest declines, and the peak age range of onset of the musculoskeletal condition changed from 35–39 to 50–54 years [5].

In Portugal, the prevalence of musculoskeletal conditions was estimated at 21.2% [6], representing the main cause of YLDs (23%) [7]. According to current guidelines, interventions for the management and treatment of these conditions should prioritize integrated, cost-effective, and high-value approaches [8], making patient-centeredness one of the cornerstones for best care [9]. Yet patient-centered care is often undervalued, under-recognized, undertrained, and considered less important than technical skills [10].

Patient-centered care has been identified as one of the priorities for improving healthcare in the 21st century, being defined as the provision of care that respects and responds to the patient’s preferences, needs, and values [11]. Specifically in physiotherapy, this model of practice involves multiple closely related aspects: individuality, communication, education, goal-setting, and support; and the social characteristics, confidence and skills, and knowledge of a patient-centered physiotherapist [12]. The therapeutic relationship has been identified as a central component of person-centered practice in physiotherapy [13], requiring presence, receptivity, genuineness, and commitment on both sides [14].

The implementation of person-centered practice has the potential to improve clinical outcomes, health-related quality of life, and cost-effectiveness of healthcare in general [15,16,17]. Regarding physiotherapy, several studies suggest that building a solid and positive therapeutic relationship is effective in the main outcomes of musculoskeletal conditions, such as pain intensity [18,19,20,21,22,23] and functional capacity [19,20,22,24,25,26]. Furthermore, its effectiveness seems to extend to other variables, namely pain self-efficacy [23,26], fear of movement [23], depression [19], general health status [19], global perceived effect [19,24], satisfaction [19], treatment adherence [27], and cost-effectiveness [28].

In recent years there has been an increasing interest in exploring how the therapeutic relationship impacts both the quality of care and clinical outcomes. In this sense, over the last 15 years, several instruments based on different conceptual structures have been developed. The most used in rehabilitation have been instruments adapted from the Working Alliance Inventory [29], based on Bordin’s model [30]. Particularly in physiotherapy, two instruments were recently developed to measure the therapeutic relationship, the Person-Centered Therapeutic Relationship in Physiotherapy Scale (PCTR-PT) [31] and the Physiotherapy Therapeutic RElationship Measure (P-TREM) [32]. The PCTR-PT was the first instrument developed to evaluate the person-centered therapeutic relationship in physiotherapy and was originally developed in Spanish. In turn, the P-TREM was built in the English language. In 2021, the Therapeutic Alliance in Physiotherapy Questionnaire-Patients was also developed to measure the therapeutic alliance in physiotherapy [33]. To date, none of these instruments have been validated for the Portuguese population.

At the time this study began, the PCTR-PT was the only instrument that had been published. Therefore, its adaptation to European Portuguese was considered a priority to respond to previously identified gaps in the Portuguese reality. The PCTR-PT presents 15 items divided into four domains [relational bond (n items = 4); individualized partnership (n items = 4); professional empowerment (n items = 3), and therapeutic communication (n items = 4)]. The response format is based on a 5 point Likert frequency scale, ranging from “strongly agree” to “strongly disagree”. The original version has adequate internal consistency (Cronbach’s alpha = 0.884) and the analysis of the intraclass correlation coefficient (ICC) revealed excellent scores (ICC = 0.900, *p* < 0.000) with 95% confidence intervals (CI) ranging from 0.846 to 0.941 [31,34].

Prioritizing person-centered practice [11], supported by a solid and positive therapeutic relationship [12,35], has the potential to improve the quality of care provided [20,36,37]. Therefore, the availability of PCTR-PT for the Portuguese population will allow patients to evaluate the established therapeutic relationship from their unique perspective [38]. Moreover, physiotherapists can use it to objectively evaluate the construction of this relationship [12]. Thus, this study aimed to cross-culturally adapt the PCTR-PT to European Portuguese and to contribute to the validation of the Portuguese version of the PCTR-PT (PCTR-PT-PV) through the study of its structural validity and internal consistency.

## 2. Materials and Methods

This study was divided into two phases: in phase 1, the cross-cultural adaptation was carried out (October 2021 to March 2022), and in phase 2, the structural validity and internal consistency of the Portuguese version of the instrument were studied (March 2022 to July 2023). The flowchart with the two phases of the study is illustrated in Figure 1.

This study received ethical approval from the Specialized Research Ethics Committee of the Escola Superior de Saúde do Instituto Politécnico de Setúbal, the Ethics Committee of the Unidade Local de Saúde de Castelo Branco and the Ethics Committee of the Unidade Local de Saúde do Norte Alentejano.

### 2.1. Phase 1: Cross-Cultural Adaptation

At this phase, a methodological study of cultural adaptation with 7 consecutive stages was carried out [39,40,41,42,43].

The translation (stage 1) began after receiving authorization from the main author of the original version of the instrument. In this stage, two translations of the instrument were produced from the original language (Spanish) to the target language (Portuguese) by two independent and bilingual translators, whose mother tongue was Portuguese. Additionally, the translators had different profiles, with one being a physiotherapist. This ensured that one of the translations promoted equivalence from a clinical perspective [39,41,42,43].

The translated synthesis version of the instrument was subsequently produced (stage 2), after discussion and consensus by a panel of four elements, independent of the translators involved in the first stage. This panel was made up of the research team, namely two students integrated into the team and two scientific advisors [39,42,43].

In stage 3, two back-translations based on the translated synthesis version developed in the previous stage were produced. For this purpose, two bilingual translators with the source language (Spanish) as their mother tongue, who were totally blind to the purpose of the translation and the original version, were recruited [41,43]. This stage assessed whether the translation was consistent, highlighting inconsistencies or conceptual errors in the process [39].

Subsequently, in stage 4, the back-translated synthesis version of the instrument and a written report were produced after consensus among the research team, a process similar to that carried out in stage 2 [39,42,43]. At the end of this stage, both synthesis versions were sent to the original authors of the instrument and their comments were integrated into the document for analysis by the panel of experts.

Then a committee of experts reviewed and consolidated the equivalence between the original and target versions (stage 5). In addition to the translators, two physiotherapists with over 10 years of experience in musculoskeletal conditions and postgraduate training related to person-centered practice and a linguist with over 10 years of experience and research in clinical linguistics were selected [39]. These criteria were verified in each expert’s submission document. At the end of the process, a pre-final Portuguese version of the instrument was developed [39,41,43].

In stage 6, a pilot study was conducted with a minimum of 30 participants to evaluate the clarity, comprehension, cultural relevance, and adequacy of the words used in the pre-final Portuguese version of the instrument (content validity) [39]. Physiotherapists who worked in two local hospitals, selected for geographic convenience, were invited to collaborate on the study, being responsible for identifying and inviting potential participants. At this stage, 44 participants were selected using non-probability convenience sampling techniques by the physiotherapists who accompanied them, according to the following inclusion criteria: (1) aged over 18-years-old; (2) have completed 3 or more physiotherapy sessions; (3) know how to read and write in Portuguese; and (4) musculoskeletal condition diagnosis. The exclusion criteria were: (1) clinical diagnosis of cardiorespiratory, neuromuscular, metabolic, genitourinary and reproductive, integumentary, mental health or vestibular condition; and (2) presence of signs or symptoms associated with serious pathology that contraindicates the practice of physiotherapy [44,45,46,47]. The participants were informed about the study and provided informed consent before participating.

In the final stage, the original authors reviewed the cross-cultural adaptation process of the instrument (stage 7). This procedure aimed to minimize the risk of bias in adapting the instrument and verify the equivalence between the original version and the new version of the instrument [43].

### 2.2. Phase 2: Psychometric Properties

At this stage, a transversal study was carried out to evaluate the structural validity and internal consistency of the preliminary Portuguese version of the instrument, following COSMIN guidelines [48].

#### 2.2.1. Sample

Participants were recruited from three local hospitals, selected for geographic convenience, using non-probability convenience sampling techniques. The eligibility criteria were the same as in phase 1. They were invited to the study and received verbal and written information about its purpose. In a subsequent session, participants who agreed to participate signed the informed consent form and filled out the instruments. To carry out the exploratory factor analysis, a minimum of 105 participants was considered, as the inclusion of seven participants per instrument item is recommended, with a minimum of 100 participants. For the study of internal consistency, a minimum of 100 is recommended [49,50].

#### 2.2.2. Instruments

Data were collected in a single moment through a characterization questionnaire and the preliminary Portuguese version of the instrument (PCTR-PT). All data were collected anonymously, as the informed consent form was not deposited in the questionnaires’ urn. The characterization questionnaire collected sociodemographic and clinical variables and was constructed based on the original studies of the instrument [31,34]. The PCTR-PT is a self-reported instrument that evaluates the person-centered therapeutic relationship in physiotherapy. The instrument construction process [31] and the detailed results of its psychometric properties [34] are described in the introductory chapter.

#### 2.2.3. Psychometric Properties

Regarding psychometric properties, structural validity was analyzed through exploratory factor analysis (EFA) using the principal components method, as this is the first validation study conducted in a culture other than the original [48,49,51]. In turn, internal consistency was estimated using Cronbach’s alpha and, additionally, by evaluating the inter-item and item-total correlation [48].

#### 2.2.4. Statistical Analysis

Statistical analysis was performed using the Statistical Package for the Social Sciences (SPSS) version 25 for the macOS Mojave operating system. In all inferential statistical tests, significance was considered when *p* < 0.05 (5% confidence interval). Descriptive statistical analyses were conducted to assess the characteristics of the participants [51].

To explore the structural validity of the instrument, the values of the Kaiser−Meyer−Olkin (KMO) test and Bartlett’s Sphericity test were initially obtained. Regarding the KMO test, it was considered that the recommendation regarding factor analysis would be: unacceptable if ≤0.5; bad, but still acceptable if ]0.5; 0.6]; mediocre if ]0.6; 0.7]; mean if ]0.7; 0.8]; good if ]0.8; 0.9]; and, excellent if ]0.9; 1.0]. As for Bartlett’s Sphericity test, it was considered that the variables would be significantly correlated if *p* < 0.001 [51]. Then the anti-image matrices for the correlations were constructed. It was considered that values of the main diagonal lower than 0.5 would indicate that the variable did not fit the structure defined by the other variables. Subsequently, the communalities were calculated, assuming that a percentage greater than 50% for all variables was a good indicator of how each variable was adequately explained by the factorial solution [51]. Then the factors that explained the correlational structure of the variables were extracted using Kaiser’s criterion, the total and each factor variance extracted, and the Scree plot’s criterion [51]. Lastly, the matrix of components was constructed to interpret the factorial solution. It was found that the factor weights of the variables in the common factors were not interpretable, since the factor weights of a reduced set of variables were not the largest possible in a single factor. This makes it impossible to attribute empirical meaning to the extracted factors. For greater definition and ease of interpretation, the factors were rotated using the Varimax method, instead of other methods like Quartimax, as there is predictably no general factor but specific factors. The critical value to determine whether the factor weight was considered significant was 0.5 [51]. After analyzing the factor weights of each item and their distribution across the retained factors, the factorial structure of the PCTR-PT-PV was determined.

To estimate internal consistency, a Cronbach’s alpha (α) coefficient between 0.70 and 0.95 was considered as appropriate. For the inter-item correlation, a value between 0.2 and 0.5 was considered appropriate, and if it was higher than 0.7, it was an indication that the items shared a very strong commonality and had little specificity. For the item-total correlation, a value greater than 0.3 was considered appropriate [50,52].

## 3. Results

### 3.1. Phase 1: Cross-Cultural Adaptation

Some differences were found between the two translation versions of the instrument (stage 1). During the first consensus panel (stage 2), the translated synthesis version of the instrument was produced, and some suggestions, such as changes in expressions or verb tenses, were recorded for further analysis by the expert panel. In turn, the back-translation stages did not present difficulties (stages 3 and 4). In general, all experts considered that the questionnaire was clear, easy to understand and answer, that the filling instructions were rigorously completed, and that the items were culturally appropriate (stage 5). During the expert panel analysis, the translation and forward meaning of the words and expressions “*paciente*” (patient), “*creo*” (believe), “*hemos conectado*” (we have connected) and “*se interesa en cómo soy como persona*” (is interested in how I am as a person) were discussed and the pre-final Portuguese version of the instrument was developed. Turning to the pilot study, this version has not changed after being analyzed and applied to 44 participants (stage 6), considering the final version was the one produced by the expert panel. Finally, the cross-cultural adaptation process was analyzed and considered successful by the original authors (stage 7) (Table 1).

### 3.2. Phase 2: Psychometric Properties

For this phase, 203 participants were recruited, and their demographic and clinical characteristics are shown in Table 2. The average age was 50 years (standard deviation (SD) [13.10]), ranging from 18 to 80 years, and 128 participants (63.1%) were female. Most participants had a partner (married or common-law married) (69.8%) and had educational qualifications equivalent to high school or higher education (68.4%). Regarding clinical characterization, the shoulder (61 participants) and the knee (52 participants) were the most frequently identified anatomical regions of diagnosis. Most participants were not taking medication (66.5%) and carried out physiotherapy sessions in the public sector (57.6%). The mean total number of sessions was 19 (minimum of 3 and maximum of 120), with a length between 45 and 60 min for most participants (109) (Table 2).

Regarding structural validity, the KMO test showed a good recommendation of homogeneity of variables (KMO = 0.869) and Bartlett’s Sphericity test concluded that the variables were significantly related (χ2 = 2840.085, *p* < 0.001). Furthermore, the values of the sampling adequacy measures in the anti-image matrices were greater than 0.5, which also supports the recommendation that the EFA could proceed. Commonality was greater than 50% for all variables, except for item 9, suggesting that it has less representation in the factorial solution obtained. As for the factor extraction, there were 3 factors with eigenvalues greater than 1 (Kaiser’s criterion) (Table 3), which explained around 75% of the total variance and more than 5% each (variance criterion) (Table 3).

In turn, the inflexion point of the curve was at the fourth factor; therefore, it also suggests a minimum of three factors (Scree plot’s criterion) (Figure 2).

By combining all criteria, it was decided in favor of a three-factor solution. After constructing the component matrix and rotating the factors using the Varimax method, it was found that no item had a factor weight > 0.5 in more than one factor, nor with a negative factor load, which suggests the maintenance of all items (Table 3). EFA resulted in a three-factor structure with 15 items: the first factor included items 9 to 15 and explained 41.8% of the variance (professional empowerment and therapeutic communication); the second factor included items 1 to 4 and explained 23.3% of the variance (relational bond); and the third factor integrated items 5 to 8, explaining 9.6% of the variance (individualized partnership).

As for internal consistency, it was considered adequate, both for the total scale (α = 0.889), and for the relational bond (α = 0.852), the individualized partnership (α = 0.889) and the professional empowerment and therapeutic communication (α = 0.953) domains. The inter-item correlation showed high correlations in the third-domain items (higher than 0.7), except for item 9, which seems to suggest that they have a very strong commonality, little specificity and that they are redundant. In turn, the item-total correlation of item 7 was 0.295, which means that this question does not seem to contribute in a relevant way to discriminate people for the construct under study. Despite this, the elimination of any item did not significantly change the result of Cronbach’s α. So, these results support the maintenance of all items of the instrument, agreeing with the results presented in the EFA.

## 4. Discussion

This study developed the European Portuguese version of the Person-Centered Therapeutic Relationship in Physiotherapy Scale, which presents a factorial structure different from the original and adequate internal consistency.

Regarding cross-cultural adaptation, the translation and back-translation stages proceeded without any major difficulties. In the pilot study, the pre-final Portuguese version of the instrument produced was not modified, ensuring the content validity of the instrument. At that stage, 44 participants were recruited with sociodemographic and clinical characteristics as those in the original validation study (n = 55), except for educational qualifications equivalent to higher education (38.6% versus 52.7%) and the diagnosis of a musculoskeletal condition (100% versus 85.3%) [34]. The mean time that participants took to complete the questionnaire was also similar between the Portuguese version (6′ 34″) and the original (6′ 40″) [34]. To date and to the best of our knowledge, this is the first study to adapt the PCTR-PT to a new culture.

To study the psychometric properties, 203 participants were recruited who also had similar characteristics to those from the original validation study (n = 366). In both studies, most participants were female; however, the proportion attending higher education was higher in the Portuguese version (36.9% versus 30.1%). In the original validation study, most participants had a diagnosis of a musculoskeletal condition (78.5%) [31]. In this study, participants had performed, on average, 19 sessions, while in the original validation study, they had performed 32 [31]. This difference can be justified by the inclusion criterion of three sessions used in the present study, as opposed to the 15 session criterion of the original validation study [31]. This inclusion criterion was decided together with the original authors, agreeing with the recommendations for the evaluation of the components of person-centered practice [38] and with the Working Alliance Inventory Rehabilitation Dutch Version criteria [53]. In both versions, participants were selected using non-probability convenience sampling techniques. Furthermore, in the Portuguese version, the recruitment centers were exclusively hospitals and fewer in number than in the original validation study (three versus nine) [34]. This methodology may suggest a possible sample selection bias, conditioning the representativeness of musculoskeletal conditions [54].

As for structural validity, in both versions, the recommendation for factor analysis was considered good and the items were significantly related (*p* < 0.001) [31]. However, in this study, three factors were extracted, as opposed to the four factors of the original validation version [31], grouping items 9 to 15 into a common factor (professional empowerment and therapeutic communication). In this sense, the factorial structure of the Portuguese version seems to support a single dimensionality between the subconstructs “professional empowerment” and “therapeutic communication”. This result may be justified by the mutual interaction between the different subconstructs related to the “therapeutic relationship” [14,55,56]. Although the presentation format of the answer options for items 9 to 15 is different to the remaining questions, both in the Portuguese version and the original [31], this difference may have contributed to the change in the factorial structure of the Portuguese version [57].

Regarding internal consistency, the PCTR-PT-PV obtained an adequate value (α = 0.889), presenting a similar value to the original version (α = 0.884). The results were also similar for the relational bond and the individualized partnership domains [31]. In the Portuguese version, the value of the professional empowerment and therapeutic communication domain was greater than 0.9 and, in the original version, this value also exceeded the threshold of 0.9 for the isolated domains of professional training and therapeutic communication [31]. These results suggest that questions 9 to 15 may be redundant in both versions [50], that is, they may potentially measure the same elements of the construct under analysis. In this case, the elimination of questions may be considered. In the Portuguese version, the study of the internal consistency was complemented with the inter-item correlation analysis, and it was found that the correlations between questions 10 and 15 were high. These findings indicate that these questions are very similar in the construct they measure, which seems to reinforce the possible redundancy and the sharing of a very strong commonality between them. This could cause an underrepresentation of the construct under analysis and lower the instrument’s reliability [58,59]. Despite this, by analyzing the item-total correlation, it was found that the elimination of any question did not significantly change the result of Cronbach’s α. In this sense, all items of the instrument were maintained, agreeing with the results of the EFA.

This study has some limitations to consider − firstly, the limitation regarding the sampling method. Participants were selected using non-probability convenience sampling by the physiotherapists who accompanied them. This method may increase selection bias, as individuals who would initially have a better therapeutic relationship may have been selected [54]. Secondly, the recruitment centers were exclusively hospitals, which may condition the representation of musculoskeletal conditions, considering the possible particularities of these locations [54]. Thirdly, the study results may be influenced by response bias, as it is consistently associated with self-report instruments [33,56]. Additionally, participants could consider that their physiotherapist’s performance was being evaluated, which could increase this bias [60,61,62,63]. Fourthly, this study did not evaluate other psychometric properties essential for making the instrument available, such as test-retest reliability, which limits its future application [48].

The results of this study constitute a relevant contribution to person-centered practice and musculoskeletal physiotherapy. This study represents an initial contribution to the availability of a quick-to-complete self-report instrument that allows the assessment of the person-centered therapeutic relationship in physiotherapy, being the first instrument adapted to European Portuguese to evaluate this outcome of interest [31,34]. The instrument can be used in other Portuguese-speaking regions; however, its use benefits from a specific cross-cultural adaptation considering different traditions, customs and particularities, to guarantee semantic, idiomatic, experiential and conceptual equivalence [39]. The research indicates that the use of an instrument that evaluates the therapeutic relationship may be of interest to patients, physiotherapists, health services, researchers, and other interested parties. This instrument may have the ability to: objectively measure the construct of therapeutic relationship [38]; provide relevant feedback on the performance of Portuguese physiotherapists and thus contribute to their learning and professional development [12,37]; explore the impact of therapeutic relationships on specific outcome measures [37] and the quality of care provided in physiotherapy [36]; and contribute to the development of guidelines in these areas of care [35,64].

## 5. Conclusions

This study was carried out in two phases, following the defined objectives: cross-culturally adapt the PCTR-PT to European Portuguese and contribute to the validation of its Portuguese version, in individuals with musculoskeletal conditions. This research contributed to the first adaptation of an instrument for the Portuguese population that allows the assessment of the person-centered therapeutic relationship in physiotherapy, which highlights the relevance of this study in person-centered practice and musculoskeletal physiotherapy in Portugal.

In conclusion, this study successfully culturally adapted a self-report instrument into European Portuguese, which allows the assessment of the person-centered therapeutic relationship in physiotherapy. Moreover, the PCTR-PT-PV has a three-factor structure and adequate internal consistency. Further research is needed to contribute to the validation of the instrument, such as assessing test-retest reliability, and to validate the adequacy of the factorial structure, to make the instrument available to the Portuguese population.

## Figures and Tables

**Figure 1 healthcare-12-02455-f001:**
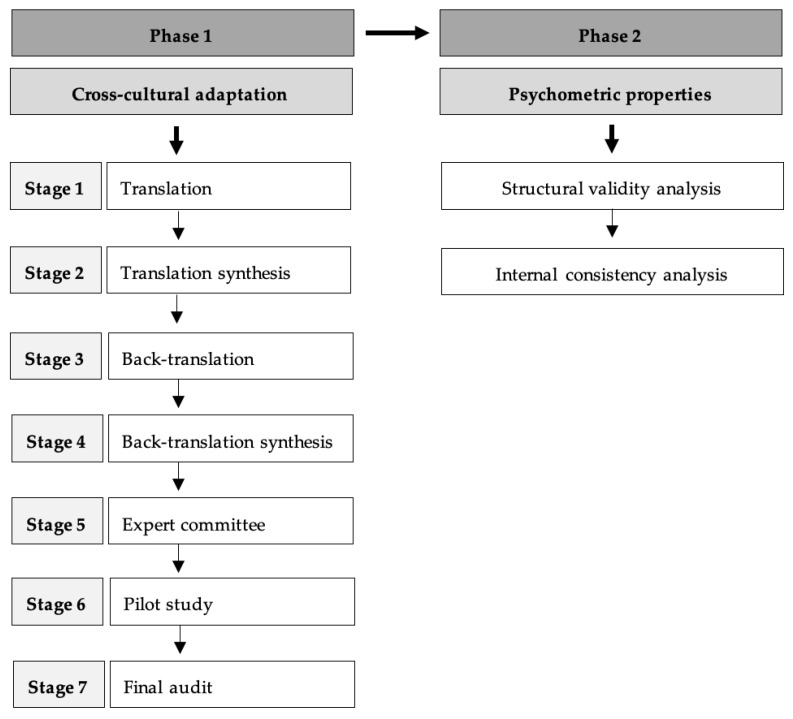
Flowchart diagram, including the stages of phase 1 and the analysis of phase 2.

**Figure 2 healthcare-12-02455-f002:**
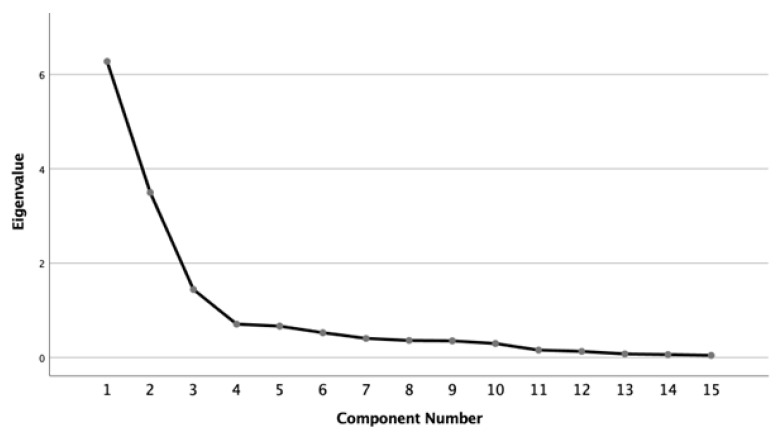
A Scree plot of the component number against eigenvalue.

**Table 1 healthcare-12-02455-t001:** Participants involved in the cross-cultural adaptation process.

Stage	Participants
Translation	2 translators (physiotherapist + Spanish language teacher)
Translation synthesis	Synthesis panel (team of 4 researchers)
Back-translation	2 back-translators (higher education teacher and Spanish-Portuguese interpreter + journalist)
Back-translation synthesis	Synthesis panel
Expert committee	7 experts (musculoskeletal expert + musculoskeletal and clinical communication expert + linguistics expert + 4 translators)
Pilot study	44 individuals with musculoskeletal conditions recruited from 3 recruitment centers
Final audit	Original authors of the Person-Centered Therapeutic Relationship in Physiotherapy Scale

**Table 2 healthcare-12-02455-t002:** Sociodemographic and clinical characteristics of the study population.

Characteristics	Participants (n = 203)	
	*n*	%
Age, mean (SD), range	50.16 (13.10), 18–80	
Sex		
Female	128	63.1
Male	75	36.9
Marital status ^1^		
Single	35	17.3
Married	110	54.5
Common-law married	31	15.3
Widowed	3	1.5
Divorced	23	11.4
Educational level		
Elementary school or inferior	23	11.4
Middle school	41	20.2
High school	64	31.5
University	75	36.9
Anatomical region of diagnosis		
Low back	38	18.7
Shoulder	61	30
Knee	52	25.6
Ankle/foot	28	13.8
Other ^2^	83	40.9
Medication		
Yes	68	33.5
No	135	66.5
Physiotherapy sector		
Public	117	57.6
Private	84	41.4
Other	2	1
Physiotherapy sessions, mean (SD), range	19.42 (20.96), 3–120	
Sessions per week		
1	1	0.5
2	53	26.1
3	40	19.7
4	2	1
5	107	52.7
Length of the sessions		
30 to 45 min	27	13.3
45 to 60 min	109	53.7
More than 60 min	67	33

^1^ Missing for one participant; ^2^ includes cervical, elbow, wrist, hip, sacroiliac, and thoracic regions.

**Table 3 healthcare-12-02455-t003:** Exploratory factor analysis using principal component analysis with Varimax rotation.

Item	Rotated Component Matrix *			Communalities
	Factor 1	Factor 2	Factor 3	
1	0.086	**0.785**	0.215	0.669
2	0.063	**0.829**	0.103	0.702
3	0.100	**0.825**	0.134	0.709
4	0.098	**0.801**	0.257	0.717
5	0.098	0.466	**0.586**	0.570
6	0.092	0.213	**0.801**	0.696
7	0.048	0.118	**0.821**	0.690
8	0.053	0.168	**0.792**	0.659
9	**0.612**	0.069	0.058	0.383
10	**0.888**	0.036	0.232	0.843
11	**0.954**	0.061	−0.001	0.913
12	**0.959**	0.117	0.001	0.934
13	**0.935**	0.079	0.057	0.884
14	**0.945**	0.120	0.121	0.922
15	**0.958**	0.078	−0.006	0.924
Eigenvalues	6.276	3.497	1.443	
Variance explained	41.8%	23.3%	9.6%	

* Bolded factor weights are greater than 0.5 in absolute value.

## Data Availability

Data described in the manuscript and the Portuguese version of the instrument will be made available upon request to the corresponding author, pending application and approval.

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
