# Peer review of "Cross-Cultural Adaptation and Validation of the Person-Centered Therapeutic Relationship in Physiotherapy Scale to European Portuguese"

_healthcare, 2024, doi:10.3390/healthcare12232455_

Round 1

Reviewer 1 Report

Comments and Suggestions for Authors

Reviewer Comments

The manuscript (healthcare-3295202) presented a cross-cultural adaptation techniques and assessment of psychometric properties, which makes it an interesting study on pateient-centered care and therapeutic associations in physiotherapy. However, there are number of points needs to be carefully addressed  by the authors. The below points should be taken into considered:

1.     Authors should give explanation regarding selection of experts for the panel in Stage 5, especially with regards to details on the specific criteria for recruitating the panels and check their suitability.

2.     Authors should provide more information regarding participant recruitment method to make sure that selection bias was avoided.

3.     The manuscript indicated that Varimax rotation is used when solutions cannot be interpreted. However, the details on criteria for non-interpretation and reason for selection of Varimax instead of other rotation methods should  be included.

4.     Authors should provide more discussion about the outcomes of internal consistency, especially where correlation of inter-item parameter is higher than 0.7.

5.     Authors should consider adding more psychometric assessments, including test-retest reliability would give more support to validation process.

6.     It is also important to provide analysis on how the instrument described in the manuscript can also be employed to other Portuguese-speaking regions.

7.     Why didn’t autors consider explorations of alternative sources of recruitment to diversify the recruitment representation.

Author Response

Dear Reviewer 1,

Response to Reviewer 1

We are deeply grateful for the time and effort you invested in reviewing our manuscript. We have carefully considered the comments and suggested revisions, and tried our best to address every one of them.

Please find the detailed point-by-point responses below and the corresponding revisions or changes in the manuscript are highlighted green.

Comment 1: Authors should give explanation regarding selection of experts for the panel in Stage 5, especially with regards to details on the specific criteria for recruitating the panels and check their suitability.

Response 1: Thank you for this suggestion. As suggested, we have added information in the manuscript about the panel recruitment criteria.

Changes made in text: Then a committee of experts reviewed and consolidated the equivalence between the original and target versions (stage 5). In addition to the translators, two physiotherapists with over 10 years of experience in musculoskeletal conditions and postgraduate training related to person-centred practice and a linguist with over 10 years of experience and research in clinical linguistics were selected [39]. These criteria were verified in each expert's submission document. At the end of the process, a pre-final Portuguese version of the instrument was developed [39,41,43]. (page 4; lines 136-142)

Comment 2: Authors should provide more information regarding participant recruitment method to make sure that selection bias was avoided.

Response 2: Thank you for pointing this out, you have raised an important point here. Selection bias had already been identified as a potential limitation of the study in the discussion section (page 9; lines 359-361), but we added information to clarify the participant recruitment process.

Changes made in text: Physiotherapists who worked in two local hospitals, selected for geographic convenience, were invited to collaborate in the study, being responsible for identifying and inviting potential participants. At this stage, 44 participants were selected using non-probabilistic convenience sampling techniques by the physiotherapists who accompanied them, according to the following inclusion criteria. (page 4; lines 145-150)

Comment 3: The manuscript indicated that Varimax rotation is used when solutions cannot be interpreted. However, the details on criteria for non-interpretation and reason for selection of Varimax instead of other rotation methods should be included.

Response 3: Thanks for the comment. The criterion for non-interpretation is not completely objective and measurable, it depends on the global analysis of the data and the possibility of interpreting them according to their values, as suggested by Marôco (2021) in his book on statistical analysis. Therefore, we have added information accordingly to make it easier to understand, and to explain the use of Varimax.

Changes made in text: Lastly, the matrix of components was constructed to interpret the factorial solution. It was found that the factor weights of the variables in the common factors were not interpretable, since the factor weights of a reduced set of variables were not the largest possible in a single factor. This makes it impossible to attribute empirical meaning to the extracted factors. For greater definition and ease of interpretation, the factors were rotated using the Varimax method, instead of other methods like Quartimax, as there is predictably no general factor but specific factors. (page 5; lines 209-215)

Comment 4: Authors should provide more discussion about the outcomes of internal consistency, especially where correlation of inter-item parameter is higher than 0.7.

Response 4: Thank you very much for the suggestion. This was a point widely discussed among the research team. In discussing the internal consistency results, we begin by identifying the implications of the high Cronbach's alpha value, both in the original version and in the Portuguese version. In the Portuguese version we also performed inter-item correlation analysis, which reinforced the implications previously assumed. We readjusted and complemented the text to try to correspond to your comment.

Changes made in text: These results suggest that questions 9 to 15 may be redundant in both versions [50], that is, they may potentially measure the same elements of the construct under analysis. In this case, the elimination of questions may be considered. In the Portuguese version, the study of the internal consistency was complemented with the inter-item correlation analysis, and it was found that the correlations between questions 10 to 15 were high. These findings indicate that these questions are very similar in the construct they measure, which seems to reinforce the possible redundancy and the sharing of a very strong commonality between them. This could cause an underrepresentation of the construct under analysis and lower the instrument’s reliability [58,59]. Despite this, by analysing the item-total correlation, it was found that the elimination of any question did not significantly change the result of Cronbach's α. In this sense, all items of the instrument were maintained, agreeing with the results of the EFA. (page 10; lines 348-359)

Comment 5: Authors should consider adding more psychometric assessments, including testretest reliability would give more support to validation process.

Response 5: Thanks for pointing this out. We agree with this comment, as we had already identified this need and limitation in the discussion (page 10; lines 369-371) and conclusion sections (page 11; lines 399-401). We are working on other psychometric properties of the instrument that we expect to explore in depth in future publications.

Comment 6: It is also important to provide analysis on how the instrument described in the manuscript can also be employed to other Portuguese-speaking regions.

Response 6: Thank you for this suggestion. The instrument can be used in other Portuguesespeaking regions, however it benefits from a cross-cultural adaptation process considering the diversity of traditions, customs, ethnicities and religions of each of these territories to guarantee its equivalence. We have added this information to the text.

Changes made in text: The instrument can be used in other Portuguese-speaking regions, however, its use benefits from a specific cross-cultural adaptation considering different traditions, customs and particularities, to guarantee semantic, idiomatic, experiential and conceptual equivalence [39]. (page 10; lines 376-380)

Comment 7: Why didn’t authors consider explorations of alternative sources of recruitment to diversify the recruitment representation.

Response 7: Thanks for pointing this out. This study was developed within the scope of master's programs (page 11; lines 421-423), therefore, the selection of hospitals was made for geographic convenience. This recruitment method and its limitation were discussed in the discussion section (page 9; lines 328-329) (page 10; lines 364-366). Despite this, the minimum number of participants for the analysis and properties studied was exceeded

Reviewer 2 Report

Comments and Suggestions for Authors

Well written manuscript with proper scientific structure and comprehensible English. Authors validated and examined Portuguese version of PCTR questionnaire. Methods were appropriate, the results were adequately described and interpreted in the discussion section. I do not see any major flaws in this manuscript. The manuscript will be suitable for publication if authors clear following issues:

·      Lines 105 – 148. A flowchart diagram would improve the clarity of the methods section. 

·      Lines 153-158. Please describe these sampling techniques. It would be perfect if authors could also describe how did they collected those questionnaires. 

Author Response

We are deeply grateful for the time and effort you invested in reviewing our manuscript. Please see the attachment.

Reviewer 3 Report

Comments and Suggestions for Authors

Review for healthcare-3295202

Authors: Moreira et al.

In this research article entitled “Cross-Cultural Adaptation and Validation of The Person-Centered Therapeutic Relationship in Physiotherapy Scale to European Portuguese”, the authors conducted a cross-cultural adaptation of the person-centered therapeutic relationship in physiotherapy scale to European Portuguese and contributed to its validation through the study of its structural validity and internal consistency.

The manuscript is well written and has logic and clear ideas. Hereafter some minor comments revealed after reviewing the manuscript.

1-       One of the great things of this manuscript is that the authors highlighted the limitations, which enriched the discussion and may give some ideas about the areas of improvement.

2-       The content of page 8, lines 260-269 should be deleted as it belongs to the journal’s template.

3-       The sentence “Within these conditions, low back pain was identified as the most prevalent”, can be supported by the following recent and relevant references; doi: 10.1007/s13205-022-03170-x and doi: 10.1016/j.vibspec.2021.103279

4-       It is recommended to provide a complete title for figure 1 instead of “Scree plot” only.

5-       English language is fine just small editing is required.

Author Response

(The authors gave the same response as above.)

Reviewer 4 Report

Comments and Suggestions for Authors

I would like to state that the study is interesting and enjoyable. I believe that if the authors fix some of the deficiencies in this beautiful study, it will be a higher quality article.

1. In the abstract section, the start and end dates of the study, the age, gender ratios of the subjects, age range and the findings section should be supported with numbers.

2. Introduction; The purpose should be explained more clearly and in a separate paragraph.

3. Method: How the sample size was calculated in Phase I and Phase II, as well as the margin of error, acceptability rate and which power analysis was used should be explained. Acceptance and rejection criteria should be stated in the samples included in the study.

4. There are too many abbreviations, making it difficult to read. My advice to the authors is to reduce abbreviations as much as possible.

5. The authors should briefly explain their contributions to the literature at the end of the Discussion.

Author Response

Dear Reviewer 4,

Response to Reviewer 4

We are deeply grateful for the time and effort you invested in reviewing our manuscript. We have carefully considered the comments and suggested revisions, and tried our best to address every one of them.

Please find the detailed point-by-point responses below and the corresponding revisions or changes in the manuscript are highlighted orange.

Comment 1: In the abstract section, the start and end dates of the study, the age, gender ratios of the subjects, age range and the findings section should be supported with numbers.

Response 1: Thank you for this suggestion. We support the abstract with numerical information (we were conditioned by the number of words in this section) and also clarify the study's duration in the methodological section's opening paragraph.

Changes made in text: Methods: This study was conducted from October 2021 to July 2023 and included two phases: a methodological study of cross-cultural adaptation (phase 1) and a transversal study to assess the psychometric properties (phase 2). (page 1; lines 18-21)

During phase 2, 203 individuals [mean age: 50,16 ± 13,10 years (range 18-80 years)] with musculoskeletal conditions, mostly female (63,1%), were recruited. (page 1; lines 24-26)

This study was divided into two phases: in phase 1, the cross-cultural adaptation was carried out (October 2021 to March 2022), and in phase 2, the structural validity and internal consistency of the Portuguese version of the instrument were studied (March 2022 to July 2023). (page 3; lines 99-102)

Comment 2: Introduction; The purpose should be explained more clearly and in a separate paragraph.

Response 2: Thanks for pointing this out. We have tried to clarify the purpose and importance of the study in its own paragraph, along with the objectives of the study.

Changes made in text: Prioritizing person-centred practice [11], supported by a solid and positive therapeutic relationship [12,35], has the potential to improve the quality of care provided [20,36,37]. Therefore, the availability of PCTR-PT for the Portuguese population will allow patients to evaluate the established therapeutic relationship from their unique perspective [38]. Moreover, physiotherapists can use it to objectively evaluate the construction of this relationship [12]. Thus, this study aimed to cross-culturally adapt the PCTR-PT to European Portuguese and to contribute to the validation of the Portuguese version of the PCTR-PT (PCTR-PT-PV) through the study of its structural validity and internal consistency. (page 2; lines 89-97)

Comment 3: Method: How the sample size was calculated in Phase I and Phase II, as well as the margin of error, acceptability rate and which power analysis was used should be explained. Acceptance and rejection criteria should be stated in the samples included in the study.

Response 3: Thanks for the comment. Considering the methodological nature of the study (adaptation and validation of an instrument), the sample size is given by the minimum number of participants for the development of a given phase or for the study of a certain psychometric property. According to the evidence consulted [39,49,50], this is the criterion for sample size, and the margin of error, acceptance rate or power analysis is not calculated. To be clearer, we added information to the manuscript regarding the minimum number of participants for the pilot study and for the analysis of exploratory factor analysis and internal consistency. The eligibility criteria for participants, which are identical for both phases, are listed in page 4, lines 150-157.

Changes made in text: In stage 6, a pilot study was conducted with a minimum of 30 participants to evaluate the clarity, comprehension, cultural relevance, and adequacy of the words used in the pre-final Portuguese version of the instrument (content validity) [39]. (page 4; lines 143-145)

To carry out the exploratory factor analysis, a minimum of 105 participants was considered, as it is recommended the inclusion of seven participants per instrument item, with a minimum of 100 participants. For the study of internal consistency a minimum of 100 is recommended [49,50]. (pages 4 and 5; lines 171-175)

Comment 4: There are too many abbreviations, making it difficult to read. My advice to the authors is to reduce abbreviations as much as possible.

Response 4: Thank you very much for the comment. We went through the entire manuscript to improve this. For example, we replaced "PCTR-PT" with "instrument" where applicable and removed single-use abbreviations from the body of the manuscript. Some examples:

- This study received ethical approval from the Specialized Research Ethics Committee of

the Escola Superior de Saúde do Instituto Politécnico de Setúbal (CEEI - ESS/IPS), the

Ethics Committee of the Unidade Local de Saúde de Castelo Branco (ULSCB) and the

Ethics Committee of the Unidade Local de Saúde do Norte Alentejano (ULSNA). to This

study received ethical approval from the Specialized Research Ethics Committee of the

Escola Superior de Saúde do Instituto Politécnico de Setúbal, the Ethics Committee of the

Unidade Local de Saúde de Castelo Branco and the Ethics Committee of the Unidade

Local de Saúde do Norte Alentejano. (page 3; lines 107-110)

- In 2021, the Therapeutic Alliance in Physiotherapy Questionnaire-Patients (CAF-P) was

also developed to measure the therapeutic alliance in physiotherapy [33]. to In 2021, the

Therapeutic Alliance in Physiotherapy Questionnaire-Patients was also developed to

measure the therapeutic alliance in physiotherapy [33]. (page 2; lines 76-77)

Comment 5: The authors should briefly explain their contributions to the literature at the end of the Discussion.

Response 5: Thank you very much for the suggestion. We did our best to optimize the information we had about the study's implications and contribution to the literature.

Changes made in text: The results of this study constitute a relevant contribution to personcentred practice and musculoskeletal physiotherapy. This study represents an initial contribution to the availability of a quick-to-complete self-report instrument that allows the assessment of the person-centred therapeutic relationship in physiotherapy, being the first instrument adapted to European Portuguese to evaluate this outcome of interest [31,34]. (page 10; lines 372-376)

The research indicates that the use of an instrument that evaluates the therapeutic relationship may be of interest to patients, physiotherapists, health services, researchers and other interested parties. This instrument may have the ability to: objectively measure the construct of therapeutic relationship [38]; provide relevant feedback on the performance of Portuguese physiotherapists and thus contribute to their learning and professional development [12,37]; explore the impact of therapeutic relationships on specific outcome measures [37] and the quality of care provided in physiotherapy [36]; as well as contribute to the development of guidelines in these areas of care [35,64]. (page 10; lines 380-387)

Round 2

Reviewer 1 Report

Comments and Suggestions for Authors

The author's have made significant changes to the manuscript. It can be considered for publication in its current form.

Reviewer 4 Report

Comments and Suggestions for Authors

It is seen that the authors made the requested changes. I think the article is better as it is. The article can be accepted as it is.